# Research on the impact effect of multimodal transport on domestic and international dual circulation: Evidence from China's railway and water transport

**Liu Wei**[1], **Zheng Xueli**[1]*, **Li Yunhan**[2,3], **Li Xia**[4], **Liu Li**[1]

**1** School of Management, Henan University of Technology, Zhengzhou, Henan, China, **2** Transportation and Economic Research Institute, China Academy of Railway Sciences Co., Ltd., Beijing, China, **3** Urban Transport and Modern Logistics Research Institute, Planning Research Institute, Ministry of Transport, Beijing, China, **4** Henan Provincial Logistics Association, Zhengzhou, Henan, China

* shirley6769@126.com

## Abstract

As an important part of the modern integrated transportation system, multimodal transport can promote changes in the transportation structure and serve as a crucial lever for upgrading the logistics industry and promoting domestic and international economic and trade development. Based on panel data from 31 provinces in China from 2017 to 2023, a fixed-effects model was constructed to empirically test the mechanism and impact of multimodal transport on domestic and international dual circulation. The findings reveal that the development of multimodal transport has a positive effect on both domestic and international circulations, with a more significant impact on the latter. Specifically, waterway multimodal transport significantly promotes both circulations, while railway multimodal transport only significantly affects the domestic circulation. Furthermore, the impact of multimodal transport on domestic and international dual circulations exhibits notable regional heterogeneity, with multimodal transport in eastern regions having a stronger effect on international circulations and that in central regions having a stronger effect on domestic circulations. This paper analyzes the theory of the multimodal transport's impact on domestic and international dual circulation, provides a practical basis and reference for further research on the development of multimodal transport in relation to domestic and international dual circulation.

## 1. Introduction

In January 2022, China released the "Work Plan for Promoting the Development of Multimodal Transport and Optimizing and Adjusting the Transportation Structure (2021–2025)", which pointed out that the goal is to accelerate the construction of a strong transportation country, focus on developing multimodal transport, and accelerate the construction of a modern comprehensive transportation system that is safe, convenient, efficient, green, and economical. Multimodal transport refers to a transportation mode where goods are loaded onto a single and unchanged transport unit, combining two or more transportation modes such as

**Data availability statement:** All data is publicly available. The original data of the article is in the Supporting Information files. The evaluation indicators for the domestic and international dual-circulation system (total sales of social retail goods and total import and export trade) and the evaluation indicator for intermodal transportation (sum of railway and waterway freight volume) are sourced from the National Bureau of Statistics of China. The control variables (regional economic development level, local fiscal expenditure on transportation, employment in railway and waterway transportation, and per capita disposable income) are primarily sourced from the "China Fiscal Yearbook," "China Statistical Yearbook," and the statistical yearbooks of various provinces.

**Funding:** Science and Technology project of Henan Provincial Department of Transport "Application of Multimodal transport in Express Logistics" (No. : 2018-2-1), "Research on the Construction of East-bound Multimodal Transport in Henan Province" (No. : 2021G1); Funded by Logistics Research Center, Key Research Base of Humanities and Social Sciences, Henan University, "Research on Policy Support System and Effect Evaluation of Multimodal Transport in China" (No. : 2020-JD-04). The funders had no role in study design, data collection and analysis, decision to publish, or preparation of the manuscript.

**Competing interests:** The authors have declared that no competing interests exist.

road, railway, waterway, and air transport organically, through coordination, transformation, and connection between different transportation modes without manipulating the goods[1]. As an essential part of the construction of a powerful transportation nation and the national comprehensive three-dimensional transportation network, multimodal transport is a significant pillar for serving the China's domestic and international dual-circulation and fostering a new development paradigm.

On April 10, 2020, at the seventh meeting of the Central Committee for Finance and Economics, China' General Secretary Xi Jinping emphasized that China should build a new development pattern with domestic circulation as the mainstay and domestic and international circulations mutually promoting each other. The domestic and international dual-circulation represents a new paradigm for China's economic development, encompassing the domestic grand cycle and the external cycle in the international market. These two cycles reinforce each other, forming an organic whole. The domestic grand cycle emphasizes leveraging the advantages of China's vast domestic market to expand domestic demand, while the domestic and international dual-circulation promotes a higher level of opening up to the outside world, making better use of both domestic and international markets and resources. This new development paradigm of dual-circulation places higher demands on multimodal transport, necessitating more efficient, convenient, and environmentally friendly transportation modes to meet the needs of emerging industries [2]. Since the China's Ministry of Transport and the National Development and Reform Commission jointly issued the "Promoting Projects for Multimodal Transport" in 2015, over a hundred multimodal transport demonstration projects have been successively declared, reviewed, and announced. Multimodal transport has provided service support for major national strategies such as China's "Belt and Road" Initiative, dual carbon goals, and a powerful transportation nation, earning strong policy support from government departments. The new paradigm of dual-circulation brings fresh ideas to the development of multimodal transport. With the implementation of the "expanding domestic demand" strategy and the steady recovery of China's economy, favorable policies for multimodal transport will continue to be intensified.

Currently, national multimodal transport demonstration routes have been opened in 28 provinces, basically covering national comprehensive transportation hub cities and the main framework of the national integrated three-dimensional transportation network. The demonstration projects that have applied for acceptance have completed investments exceeding RMB 20 billion, driving over 1,000 upstream and downstream enterprises to participate in multimodal transport-related work, providing crucial support for smooth domestic and international economic circulations. Positive breakthroughs have been made in aspects such as the legalization of property rights in railway bills of lading, information interconnection and sharing, and integrated customs clearance. In 2022, the demonstration projects collectively completed around 7.2 million TEUs (Twenty-foot Equivalent Units) of container multimodal transport. Compared to road transport, this reduced logistics costs by over RMB 10 billion, making significant contributions to optimizing and adjusting the transportation structure and winning the "Blue Sky Defense Battle." The new development paradigm of dual-circulation provides broader development space and opportunities for multimodal transport, driving it towards higher-quality development. At the same time, as an essential part of the construction of a powerful transportation nation and the national integrated three-dimensional transportation network, multimodal transport serves as an important pillar for serving the domestic and international dual-circulation and fostering a new development paradigm. By optimizing transportation routes, improving transportation efficiency, and reducing transportation costs, multimodal transport provides better and more efficient logistics services for the dual-circulation, facilitating the smooth operation of both domestic and international circulations.

It is evident that multimodal transport and the dual-circulation are closely related, mutually promoting each other's development.

Most of the existing literature focuses on research areas such as multimodal transport route optimization [3–4], network design, "one-stop" system, and effect evaluation. Specifically, regarding the evaluation of multimodal transport development effects, existing studies have analyzed its impact on driving domestic regional economic growth or promoting international trade. However, there is a lack of research examining its dual impact on both domestic and international circulations from the perspective of the dual-circulation paradigm. Additionally, while some scholars have sporadically studied the operational effects of individual multimodal transport corridors or specific modes like railway, waterway, and aviation, there is limited comparison of the varying economic promotion effects among different types of multimodal transport. Consequently, what is the mechanism behind multimodal transport empowering the domestic and international dual-circulation? Can it simultaneously boost both the domestic and international circulations? Do the promotion effects vary among different regions and multimodal transport modes? These questions offer significant room for further exploration. In light of this, the marginal contribution of this paper lies in exploring the mechanism of multimodal transport's role in domestic and international dual-circulation. By constructing empirical models to analyze the impact of multimodal transport on both the domestic and international circulations separately, this paper contributes to uncovering the differences in multimodal transport's promotion effects across various dimensions of the dual-circulation, thereby providing a reference for optimizing multimodal transport support policies.

The rest of this paper is structured as follows: Section 2 summarizes the existing literature on multimodal transport and domestic and international dual-circulation. Section 3 conducts a theoretical analysis to explain how multimodal transport affects domestic and international dual-circulation from various aspects and proposes hypotheses for this paper. Section 4 involves model construction and data selection. Section 5 presents empirical analysis, heterogeneity analysis, and robustness tests. Finally, Section 6 summarizes the research conclusions and puts forward policy recommendations.

## 2. Literature review

Existing literature mainly focuses on the research of multimodal transport route optimization, network design, "intermodal one-bill coverage mechanism", and effect evaluation. However, there are relatively few studies examining the dual impact on both the internal and external regions from the dual perspectives of regional economic internal activity and openness to the outside world.

### 2.1. Multimodal transport network design

Zhang Dezhi et al [11] studied the design optimization and subsidy models of water-land intermodal transportation networks from a low-carbon perspective, considering the interactive game behavior between government regulatory departments and logistics users, and constructed a water-land intermodal transportation logistics network optimization model based on bilevel programming. Hou Yujie et al [12] proposed a new sustainable transportation mode, the underground logistics system, to address common issues faced by current port city development. Wu Peng et al [13] constructed a multi-objective mixed-integer nonlinear programming model under different carbon emission policies and transformed it into an equivalent linear model based on the problem's characteristics. Experimental results showed that compared with traditional genetic algorithms and the commercial solver Lingo, the

improved adaptive genetic algorithm could obtain more satisfactory solutions. Based on the multi-objective Pareto optimization idea, Ai Ziyan et al [14] designed a fast non-dominated sorting genetic algorithm to solve the optimal transport service scheme, and the research results can provide freight owners with different transport service optimization schemes that meet their needs. Zhang Guanxiang et al [15] showed that the network risk and transportation cost decreased with the increase of the number of transit points, proving that the tiered charging strategy of transit points can effectively regulate the network risk of dangerous goods multimodal transport.

## 2.2. Multimodal transport route optimization

Bontekoning et al [5] regarded route optimization as an emerging and significant area within transportation research. Janic [6] developed a multimodal transportation route optimization model with time windows to minimize total costs. Liu et al [7] investigated a multimodal transportation multi-objective optimization problem jointly solved by single-objective and multi-objective genetic algorithms through model decomposition. Focusing on profit maximization, Ji et al [8] established an optimization model for sea-rail container operations and designed a heuristic algorithm to obtain optimal railway transportation routes and service pricing. Hu Zuo'an et al [9] suggested that multimodal transportation decision-makers should predict the impact of uncertain factors, select appropriate maximum regret values, and pay attention to node mixed time window constraints to reduce costs and improve efficiency. Yang Zhe et al [10] established a low-carbon, low-cost multimodal transportation route optimization model under fuzzy demand and fuzzy transportation time. To address the issue that continuous meta-heuristic algorithms cannot directly solve discrete combinatorial optimization models, they designed a priority-based general encoding method.

## 2.3 Multimodal transport one-bill coverage mechanism

At present, there are many studies on the "intermodal one-bill coverage mechanism" of multimodal transport. Zhao Yu [16] analyzed the current development status of multimodal transportation in China and the issues surrounding the promotion of the "one-document" system, proposing the advantages and main measures for promoting the "one-document" system for multimodal transportation with railway transportation as the core. Xian Jinghan [17] took the multimodal transportation document rules as a blueprint, reviewed and compared the application of multimodal transportation and transportation documents based on existing international cargo transportation rules, ensuring the feasibility of multimodal transportation document rules that safeguard multimodal transportation document functions and clarify their legal nature. Zhu Liangyong et al [18], combining Petri net theory, proposed establishing a multimodal transportation process model and optimization model using sea-rail intermodal transportation as an example, and suggested constructing an information sharing platform based on blockchain for the multimodal transportation "one-document" system. Based on two aspects of intermodal transport development environment and the organization of train operation, Wei Zhen et al [19] deeply expounded the development status of Guangxi railway multimodal transport, analyzed the existing problems, and put forward countermeasures for the development of Guangxi railway multimodal transport. On the basis of describing the general situation of "one single system" of multimodal transport in China, Shen Bing et al [20] put forward the construction strategy of "one single system" of multimodal transport in China in view of the existing problems of "one single system" of multimodal transport in China, taking "one single system" of multimodal transport as the main line of business.

### 2.4. Evaluation of the development effects of multimodal transportation

Specifically regarding the evaluation of the development effects of multimodal transportation, Xi Yue [21] analyzed the impact of multimodal transportation on domestic regional economic growth and believed that multimodal transportation is a new driving force for regional economic development. Some scholars have also sporadically studied specific multimodal transportation corridors. Zhang Ao et al [22] took the Yangtze River Delta region as an example to explore the impact of coordinated multimodal transportation development on promoting the establishment of favorable circulation conditions in the region. Li Lingxiang et al [23], through the comparison and selection of transportation modes and the conduct of project economic evaluation analysis, elaborated on its financial feasibility and implemented the concept of enterprise service guarantee requirements. Starting from the definition of the concept of multimodal transport, Li Chunhua et al [24] carried out a theoretical traceability of the multimodal transport system. On this basis, they thoroughly analyzed the development status of multimodal transport from the aspects of the promotion of national policies and the development track of various modes of transport, and explored four types of operation modes of multimodal transport. Based on this, they conducted a comprehensive assessment of the effects of multimodal transport under the COVID-19 epidemic. The paper reveals the existing problems and their underlying reasons, and finally designs the "three in one" strategy for the optimization of multimodal transport in Chinese logistics enterprises from three levels: intelligent multimodal transport platform (government), multimodal transport enterprise alliance (industry) and multimodal transport mode combination (enterprise).

The marginal contribution are as follows. Firstly, existing research has mainly focused on optimizing multimodal transport routes, network design, and one-bill coverage mechanism. However, there are also a few literature evaluations of the economic effects of multimodal transport, and the significance and role of multimodal transport are still not understood. Secondly, this paper analyzes the influence mechanism of multimodal transport enabling domestic and foreign double cycles, and expounds whether multimodal transport can promote domestic great cycle and international external cycle at the same time. Thirdly, through the heterogeneity analysis, analyze whether there are differences in the impact effects of different regions and intermodal modes, and provide policy references for promoting the development of multimodal transportation in different regions.

### 3. Theoretical analysis and research hypothesis

Multimodal transport empowers the domestic circulation by promoting economic development. (1) Multimodal transport reduces social logistics costs and promotes the construction of a unified domestic market and the formation of a new development paradigm. The development of multimodal transport is one of the effective means to reduce logistics costs in China, which can effectively promote the cost reduction of the logistics industry and improve the efficiency of commodity circulation. Multimodal transport innovates the transportation organization mode, simplifies the transportation process, establishes a more perfect full-process service system, and creates a healthy and open market environment. Rapid development of multimodal transport can effectively save resource allocation costs, promote the construction of a unified domestic market, and facilitate the formation of a new development paradigm. Through the development of multimodal transportation, various regions have connected channels for different modes of transportation, built information sharing platforms, and achieved information resource sharing. At the same time, improving infrastructure connectivity and strengthening the construction of multimodal transport backbone channels can effectively promote the rapid growth of domestic trade volume. (2) Accelerating the integration of

factor resources and promoting the clustering of hub economies and advantageous industries. The hub economy is a large-scale industrial development model that utilizes economic factor resource aggregation platforms (transportation hubs, logistics hubs, logistics service platforms, financial platforms, etc.) to aggregate, diffuse, and channelize business flows, logistics flows, capital flows, information flows, and passenger flows [21]. Economic flows are not limited by geographical location or spatial development. Industrial platforms such as e-commerce, logistics distribution centers, trade settlement centers, and call service centers can diffuse through international and domestic services, which are important avenues for developing hub economies. A high-density logistics activity cluster will form around the hub, where logistics enterprises provide various logistics services, creating two major economic driving effects: on the one hand, the highly developed producer services focused on logistics activities; on the other hand, manufacturing and trade enterprises have a high sensitivity to logistics timeliness and costs and will arrange their industrial layouts around the hub. As a result, multimodal transport hubs become an economic engine, fostering the hub economy. Many early port-related industries in China gradually formed around transfer hubs, bringing substantial distribution and transshipment business, which in turn promoted the economic development of coastal cities and ports.

Based on this, Hypothesis 1 is proposed

H1: Intermodal transportation significantly promotes the development of the domestic dual-circulation economy.

Multimodal transport empowers the international external circulation by expanding opening-up. (1) Expanding the scope of opening-up promotes the level of opening-up in inland areas [22]. The core component of the "the Belt and Road" is the establishment of closer economic and trade ties with countries along the land route [23]. Over the past 40 years of reform and opening-up in China, international trade has primarily relied on sea transportation, leading to rapid economic development concentrated mainly in coastal regions. The economic development of central and western regions, dominated by inland cities, has been relatively slow and still requires further opening-up for new development. Among the many countries connected by land routes in western and southwestern China, multimodal transport offers cost advantages in terms of efficiency and operational models compared to sea transportation [25], bringing more development opportunities for the opening-up of inland regions in central and western China. Taking Henan Province, an inland region, as an example, it has focused on institutional innovation to accelerate the construction of an interconnected, globally logistics-oriented, and end-to-end multimodal transport service system, pioneering a new path for multimodal transport development in inland provinces and being listed as one of the first pilot regions for "Transportation Power" by the Ministry of Transport. (2) Unleashing the radiation effect of major logistics corridors accelerates international capacity cooperation. Promoting multimodal transport helps to target both domestic and international markets, leveraging high-quality inward and high-level outward efforts to innovate cooperation models, optimize the global industrial layout, advance international capacity cooperation, embed into global industrial and supply chains, and enhance the level of foreign trade development [26]. Multimodal transport can strengthen cooperation with countries along the "the Belt and Road", promote the development of cross-border logistics and supply chains, and facilitate international trade and investment. Taking Shaanxi Province in China as an example, leveraging its geographical advantages, it has continuously improved its comprehensive and three-dimensional transportation network, accelerated the construction of the Xi'an Assembly Center for China Railway Express, and built an important opening-up corridor facing Central Asia, South Asia, and West Asia. In 2022, 306 enterprises in Shaanxi Province achieved a cumulative

foreign investment of 6.88 billion dollars, including 1.82 billion dollars in countries and regions along the "the Belt and Road".

Based on this, Hypothesis 2 is proposed

H2: Intermodal transportation significantly promotes the development of the international external circulation.

The mechanism analysis of multimodal transport enabling domestic and international double cycle is shown in Fig 1:

## 4. Methods and data

### 4.1. Model specification

For panel data processing, we generally choose fixed effects or random effects for regression. At this time, Hausman test is needed to determine whether individual effects are related to explanatory variables, so as to decide whether to use fixed effects model or random effects model. According to the Hausman test, rejecting the null hypothesis indicates that the individual effect is related to the explanatory variable, and the fixed effect model is more appropriate in this case. Using a fixed-effects regression model, we analyze the specific impacts of multimodal transport development on domestic and international dual-circulation systems, and establish the following detailed model:

$$Yit = \alpha 0 + \alpha 1 Xit + \alpha 2 Controlsit + \sum pro + \sum year + \varepsilon it \tag{1}$$

Wherein, $Y_{it}$ is the explained variable, representing the development level of domestic and international dual-circulation in $i$ provinces and $t$ years; $\alpha 0$ is the intercept term; $\alpha 1$, $\alpha 2$ are coefficients, represent the specific impacts of multimodal transport on domestic and international dual-circulation; $X_{it}$ is the core explanatory variable, represents the level of multimodal transport in different provinces and years; $Controlsit$ represents control variables, including regional economic development level, government support, employment population, and per capita disposable income; $pro$ represents the provincial effect, $year$ represents the year effect, and $\varepsilon it$ represents the error term.

The dependent variables of domestic large cycle and international external cycle change with the change of multimodal transport and time, and the statistical data are in units of years. Is the intercept term, $\alpha 0$ is the conditional average when $X_{it} = 0$, that is, $Y_{it}$, the development level of domestic and international double circulation when the level of multimodal transport

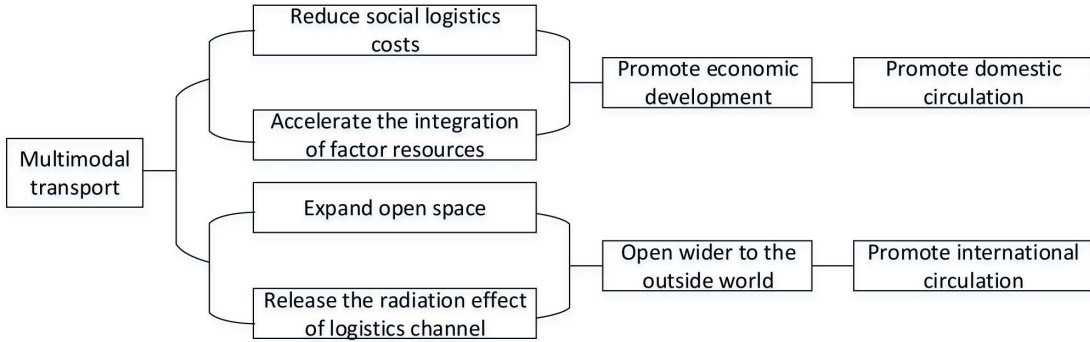

**Fig 1. Mechanism analysis diagram.**

is 0. For the purpose of observation, $\alpha 1$ represents the impact of multimodal transport on the domestic and international double cycle.

## 4.2. Variable selection

### 4.2.1. Dependent variable.
Based on the availability and scientificity of data, and referring to existing research literature, the total sales of social retail goods and the total import and export trade are selected as evaluation indicators for domestic and international circulations, respectively [27,28]. To develop the domestic cycle, it is necessary to ensure smooth production, distribution, circulation, and consumption processes in the domestic market, with smooth consumption being the driving force for other links. Therefore, this paper selects the total sales of social retail goods as the measurement indicator for the domestic cycle. Developing international circulation, advocating that the construction of external transportation network can effectively expand the scale of the global market, promote the deepening of "globalization" through the "Belt and Road", and further open China's coastal areas, with the purpose of absorbing surplus labor on the one hand and introducing foreign investment to develop the economy on the other. Furthermore, the entropy method is adopted to synthesize the domestic and international cycles into a single indicator, reflecting the development level of the domestic and international dual-circulation system.

### 4.2.2. Independent variable.
Taking "intermodal transportation" as the core explanatory variable, the sum of railway and waterway freight volume is selected as the evaluation indicator reflecting the level of intermodal transportation in a region (province), in order to study the specific impact of intermodal transportation on the domestic and international dual-circulation system [25].

### 4.2.3. Control variables.
The following indicators are selected as control variables: regional economic development level, local fiscal expenditure on transportation, employment in railway and waterway transportation, and per capita disposable income. (1) Regional Economic Development Level (REDL): Represented by per capita GDP, which is one of the key indicators to measure the level of regional economic development. A higher per capita GDP in a region indicates a higher level of economic development. (2) Government Support (GS): Represented by local fiscal expenditure on transportation, which can better measure the support of local governments for the local transportation industry. The level of financial investment in different regions determines the construction of intermodal transportation infrastructure and equipment, which in turn affects the regional competitiveness of intermodal transportation. (3) Railway and Waterway Transportation Employment (RWTE): Represented by the sum of employment in railway and waterway transportation. (4) Per Capita Disposable Income (PCDI): Represented by per capita disposable income of residents. Additionally, during the empirical analysis, all control variables are log-transformed to eliminate the influence of dimensionality.

### 4.2.4. Data sources and statistical descriptions.
Panel data from 31 provinces in China from 2017 to 2023 are selected as the research samples, a small amount of missing data was completed by interpolation or mean method. Among them, the evaluation indicators for the domestic and international dual-circulation system (total sales of social retail goods and total import and export trade) and the evaluation indicator for intermodal transportation (sum of railway and waterway freight volume) are sourced from the National Bureau of Statistics of China. The control variables (regional economic development level, local fiscal expenditure on transportation, employment in railway and waterway transportation, and per capita disposable income) are primarily sourced from the "China Fiscal Yearbook," "China Statistical

Yearbook," and the statistical yearbooks of various provinces. The statistical descriptions of the main variables are shown in Table 1.

## 5. Empirical analysis

### 5.1 Benchmark regression analysis

To empirically test the effects of multimodal transport on domestic and international circulations, a benchmark regression using fixed-effects was conducted separately, the results shown in Tables 2 and 3.

Model 1 in Table 2 considers only the impact of multimodal transport on the domestic circulation. The regression results indicate that, the coefficient is positive and the P-value is less than 0.01, the development of multimodal transport has a significant positive effect on the domestic circulation. By adding control variables to Model 1, based on the regression results of Model 2, it can be seen that after including the control variables, the impact of multimodal transport on the domestic circulation remains positive and significant, with a regression coefficient of 0.066. This shows that the regression results are robust and reliable.

In Model 3 of Table 3, which only considers the impact of multimodal transport on the international circulation, the regression results indicate that, the coefficient is positive and the P-value is less than 0.01, multimodal transport has a significant positive effect on the development of the international circulation, with a regression coefficient of 0.358. After adding control variables, the regression results are shown in Model 4, and multimodal transport still has a significant positive effect on the development of the international circulation, demonstrating the robustness of the regression results.

In summary, the development of multimodal transport has diversified the modes of transportation, providing basic guarantees for domestic trade and international economic

**Table 1. Statistical description of the variable.**

| variable | index | maximum | minimum | average |
|---|---|---|---|---|
| Domestic and international Double circulation | Total sales of retail goods (100 million yuan) | 47494.90 | 618.80 | 13270.87 |
| | Total import and export trade (US$ 100 million) | 12795.70 | 3.11 | 1669.02 |
| Multimodal transport | Rail and waterway freight traffic (10,000 tons) | 227838.00 | 52.00 | 40728.62 |
| Control variables | Regional economic development level (RMB/person) | 200278.00 | 29103.00 | 73559.44 |
| | Government support (100 million yuan) | 23037.00 | 66.34 | 447.32 |
| | Number of Employed Persons | 154604.00 | 36.00 | 70269.49 |
| | Per capita disposable income (RMB) | 84834.00 | 15457.00 | 32440.19 |

**Table 2. The impact of multimodal transport on the domestic circulation.**

| variable | model 1 | | model 2 | |
|---|---|---|---|---|
| | coefficient | T | coefficient | t |
| Multimodal transport | 0.222*** | 7.30 | 0.045* | 1.90 |
| The level of regional economic development | | | 0.441*** | 6.24 |
| Government support | | | -0.030** | -2.24 |
| Number of people employed | | | -0.011 | -0.78 |
| Disposable income per capita | | | 0.031 | 1.46 |
| constant | 6.875 | 22.83 | 3.923 | 5.60 |

Note: ***, **, and *indicate significant at the levels of 1%, 5%, and 10%, respectively.

**Table 3. The impact of multimodal transport on the international circulation.**

| variable | model 3 | | model 4 | |
|---|---|---|---|---|
| | coefficient | t | coefficient | t |
| Multimodal transport | 0.550*** | 7.48 | 0.183*** | 2.76 |
| The level of regional economic development | | | 1.163*** | 4.32 |
| Government support | | | -0.053 | -1.38 |
| Number of people employed | | | -0.071* | -1.78 |
| Disposable income per capita | | | -0.067 | -0.25 |
| constant | 12.342 | 16.98 | 4.802 | 4.76 |

Note: ***, **, and *indicate significant at the levels of 1%, 5%, and 10%, respectively.

cooperation. Meanwhile, multimodal transport has also achieved the goal of reducing costs and increasing efficiency. Mixed transportation of multiple modes enables enterprises to choose modes of transportation with lower costs and higher efficiency, which will strengthen cooperation among enterprises in different regions and countries and further promote the development of international trade.

Furthermore, by comparing the differences in the promotion effects of multimodal transport on domestic and international economic cycles, it is found that for every 1% increase in the development level of multimodal transport, the domestic economic cycle will be promoted by 0.066%, while the international economic cycle will be promoted by 0.229%. This indicates that the positive effect of multimodal transport development on the international economic cycle is higher than that on the domestic economic cycle. The possible reason is that the international economic cycle relies more heavily on multimodal transport, or in other words, the better the development foundation of multimodal transport in a region, the greater the development space and first-mover advantage for the international economic cycle. In contrast, for the domestic economic cycle, a significant proportion of freight transport still relies on road transport, so multimodal transport has a less significant driving effect on the domestic economic cycle than on the international economic cycle.

## 5.2. Dimensional regression analysis

In order to compare the effects of different multimodal transport modes on the double cycle, two types of multimodal transport, rail and water, are selected here and panel data regression models are established respectively.

According to the regression results in Table 4, railway multimodal transport has a significant promoting effect on the domestic economic cycle, with a regression coefficient of 1.795, while its impact on the international economic cycle is not significant. The possible reasons are as follows: Railway transport is primarily suitable for freight transport between neighbouring countries. For transportation across several countries or over long distances, the time and cost of railway transport may not be advantageous, resulting in relatively fewer international goods transported by rail. In addition, some countries may lack comprehensive railway infrastructure, such as sparse railway networks and inconsistent track standards, which may lead to a decrease in the efficiency and reliability of international railway transportation, thereby affecting the proportion of railways in international freight transport.

As indicated by the regression results in Table 5, waterway multimodal transport has significantly contributed to both the domestic and international economic cycles. Waterway transportation, characterized by its large carrying capacity, low cost, strong adaptability, high safety, strong sustainability, extensive coverage, and minimal impact from climate, has

**Table 4. The influence of railway multimodal transport on domestic and international double cycle.**

| variable | domestic circulation | | international circulation | |
|---|---|---|---|---|
| | coefficient | t | coefficient | t |
| Railway multimodal transport | 0.113*** | 2.96 | -0.050 | -0.46 |
| The level of regional economic development | 0.002 | 1.62 | 1.260*** | 4.58 |
| Government support | -0.040*** | -2.61 | -0.047 | -1.17 |
| Number of people employed | -0.005 | -0.33 | -0.072* | -1.77 |
| Disposable income per capita | 0.001*** | 3.31 | 0.007 | 0.02 |
| constant | 7.881 | 23.26 | 5.181 | 4.98 |

Note: ***, **, and *indicate significant at the levels of 1%, 5%, and 10%, respectively.

**Table 5. The impact of waterway multimodal transport on domestic and international dual circulation.**

| variable | domestic circulation | | international circulation | |
|---|---|---|---|---|
| | coefficient | t | coefficient | t |
| Waterway multimodal transport | 0.028*** | 3.29 | 0.102*** | 5.34 |
| The level of regional economic development | 0.505*** | 5.54 | 1.543*** | 7.56 |
| Government support | -0.020* | -1.69 | -0.025 | -0.95 |
| Number of people employed | -0.035 | -1.52 | 0.110** | 2.10 |
| Disposable income per capita | 0.198** | 2.24 | -0.069 | -0.35 |
| constant | 2.021 | 4.25 | -0.121 | -0.11 |

Note: ***, **, and *indicate significant at the levels of 1%, 5%, and 10%, respectively.

emerged as one of the primary modes of both international and domestic transportation. In comparison, the promoting effect of waterway multimodal transport on the international economic cycle (with a regression coefficient of 1.680) is more pronounced than that on the domestic economic cycle (with a regression coefficient of 0.618).

## 5.3. Robustness test

To ensure the reliability of the model and regression results, it is necessary to conduct robustness tests on the actual outcomes. Drawing on the methodologies of relevant scholars [29,30], choose to perform robustness tests through stepwise regression, model substitution, and Winsorized regression.

When regressing fixed-effects models, stepwise regression is used, which involves gradually adding control variables during the regression process and observing whether there will be significant changes in the regression results. As evident from Tables 2 and 3, the regression results for multimodal transportation's impact on the domestic and international circulations are significant without including control variables. After incorporating control variables, the regression results remain significant, with minimal changes in the coefficients of influence. This indicates that the regression results in this study are relatively robust.

To further validate the robustness of the regression results, the fixed effects model is replaced by random-effects model. The results are presented in Tables 6 and 7. Comparing these results with those from the fixed-effects regression, we find that the regression outcomes remain significant even after switching to the random-effects model. Moreover, after gradually adding control variables, the results are largely consistent with those from the fixed-effects regression. This shows that the positive and significant effects of multimodal transportation on both domestic and international circulations are fundamentally robust.

**Table 6. Random-effect regression of multimodal transport on domestic circulation.**

| variable | model 5 | | model 6 | |
|---|---|---|---|---|
| | coefficient | z | coefficient | z |
| Multimodal transport | 0.240*** | 8.23 | 0.081*** | 3.28 |
| The level of regional economic development | | | 0.252 | 2.34 |
| Government support | | | -0.026* | -1.69 |
| Number of people employed | | | -0.010 | -0.60 |
| Disposable income per capita | | | 0.275** | 2.51 |
| constant | 0.743 | 4.67 | 2.869 | 6.89 |

Note: ***, **, and *indicate significant at the levels of 1%, 5%, and 10%, respectively.

**Table 7. Random-effects regression of multimodal transport on international circulation.**

| variable | model 7 | | model 8 | |
|---|---|---|---|---|
| | coefficient | z | coefficient | z |
| Multimodal transport | 0.568*** | 8.44 | 0.296*** | 4.83 |
| The level of regional economic development | | | 1.225*** | 4.20 |
| Government support | | | -0.039 | -0.94 |
| Number of people employed | | | -0.026 | -0.61 |
| Disposable income per capita | | | -0.156 | -0.52 |
| constant | 0.139 | 0.35 | 3.355 | 3.05 |

Note: ***, **, and *indicate significant at the levels of 1%, 5%, and 10%, respectively.

$$Yit = \beta0 + \beta1 Xit + \beta2 Controlsit + \tau it \qquad (2)$$

Wherein, $Y_{it}$ is the explained variable, representing the development level of domestic and international dual-circulation in $i$ provinces and $t$ years; $\beta0$ is the intercept term; $\beta1$, $\beta2$ are coefficients, represent the specific impacts of multimodal transport on domestic and international dual-circulation; $X_{it}$ is the core explanatory variable, represents the level of multimodal transport in different provinces and years; $Controlsit$ represents control variables, including regional economic development level, government support, employment population, and per capita disposable income; $\tau it$ represents the random error term.

To reduce the error caused by outliers in the benchmark regression results, a 2.5% tail reduction was applied to the dependent and core explanatory variables to eliminate extreme data in the sample. The regression results are shown in Table 8. According to the regression results of the truncated data, the impact of multimodal transport on domestic and international dual circulation is still significantly positive, indicating that the regression results are robust and reliable.

In addition, take into account the issue of endogeneity of variables in data processing. The paper proposes the following approach.

First, concerning errors arising from the original data, all research data were sourced from authoritative databases such as the National Bureau of Statistics website, China Rural Statistical Yearbook, China Fiscal Yearbook, China Statistical Yearbook, and provincial statistical yearbooks. The paper has processed missing, omitted, or abnormal raw data to substantially eliminate the impact of such errors on the regression results.

Second, addressing the influence of omitted important variables, during the regression process, we selected regional economic development level, local fiscal transportation expenditure,

**Table 8. Tail regression.**

| variable | domestic circulation | | international circulation | |
|---|---|---|---|---|
| | coefficient | t | coefficient | t |
| Multimodal transport | 0.056** | 2.08 | 0.180*** | 2.57 |
| The level of regional economic development | 0.431*** | 6.13 | 1.143*** | 4.63 |
| Government support | -0.034** | -2.51 | -0.051 | -1.46 |
| Number of people employed | -0.039*** | -2.85 | -0.044 | -1.20 |
| Disposable income per capita | 0.003 | 1.25 | -0.048 | -0.19 |

Note: ***, **, and * indicate significant at the levels of 1%, 5%, and 10%, respectively.

employment in the railway and waterway transport industry, and per capita disposable income as control variables. Employing the fixed-effects model for estimation can control for individual heterogeneity, which refers to omitted variables in the error term that do not vary over time.

Third, regarding the potential bidirectional causality between variables, the domestic and international dual-circulation paradigm represents China's new development paradigm, a national strategy aimed at promoting high-quality economic development. It serves as an objective. In contrast, multimodal transportation is a means to reform traditional transportation modes, adjusting transportation structures, enhancing efficiency, and promoting economic growth. Therefore, in the causal relationship between the two, multimodal transportation is the cause, while the domestic and international dual-circulation paradigm is the effect.

## 5.4. Heterogeneity analysis

Given the differences in economic conditions, resource endowments, and other developmental levels across regions, the impact of multimodal transportation on domestic and international dual-circulation may exhibit regional variations. Therefore, the research samples were divided into eastern, central, and western regions according to the documents of the China National Development and Reform Commission to examine whether there are regional heterogeneities in the influence of multimodal transportation development on domestic and international dual-circulation(Table 9). The results of this examination are presented in Table 10. Among them, the division of East, central and western regions is based on the social and economic development of different regions in China, and the specific distribution is shown in Fig 2.

As evident from Table 10, in China's eastern and central regions, the development of multimodal transportation has produced significant positive effects on both domestic and international dual-circulation. However, in the western region, the impact of multimodal transportation on domestic and international dual-circulation is not significant. This may be attributed to the fact that eastern China was the first region to implement the coastal opening-up policy, resulting in a relatively high overall economic development level, abundant talent resources, and numerous coastal cities with well-developed water transportation, leading to a leading position in multimodal transportation development. The central region, as an economically less developed area, also boasts a relatively developed transportation industry dominated by highways. Although multimodal transportation promotes dual-circulation, its impact is slightly lower than that in the eastern region. The western region, as an economically underdeveloped area, lags behind in economic development compared to the eastern and central regions, suffers from a shortage of human resources, and faces relatively

**Table 9. Distribution table of provinces in China's three major economic regions.**

| Eastern region | **Beijing, Fujian, Guangdong, Hainan, Hebei, Jiangsu, Liaoning, Shandong, Shanghai, Tianjin and Zhejiang** |
|---|---|
| Central region | Anhui, Heilongjiang, Henan, Hubei, Hunan, Jiangxi, Jilin and Shanxi |
| Western region | Gansu, Guangxi, Chongqing, Guizhou, Inner Mongolia, Ningxia, Qinghai, Shaanxi, Sichuan, Xinjiang, Yunnan, Tibet |

**Table 10. Heterogeneity analysis.**

| region | variable | domestic circulation | | international circulation | |
|---|---|---|---|---|---|
| | | coefficient | t | coefficient | t |
| Eastern region | Multimodal transport | 0.169*** | 4.57 | 0.275*** | 3.96 |
| | constant | 3.066*** | 3.47 | 3.354** | 2.02 |
| Central Region | Multimodal transport | 0.023*** | 2.47 | 0.124* | 1.85 |
| | constant | 2.574*** | 2.43 | 0.178 | 0.11 |
| Western region | Multimodal transport | -0.047 | -1.38 | 0.087 | 0.36 |
| | constant | -4.130*** | -5.18 | 5.850*** | 3.15 |

Note: ***, **, and *indicate significant at the levels of 1%, 5%, and 10%, respectively.

high transportation costs. Consequently, the role of multimodal transportation in promoting dual-circulation is not apparent.

Furthermore, by comparing the regression coefficients, we observe that the effects of multimodal transportation on domestic circulation are broadly similar in both the eastern and central regions, but the eastern region exerts a greater influence on international circulation compared to the central region. Within regions, multimodal transportation in the eastern coastal areas has a stronger promotional effect on international circulation (regression coefficient of 0.466), significantly higher than its impact on domestic circulation (regression coefficient of 0.170). Conversely, in the central region, multimodal transportation exhibits a stronger promotional effect on domestic circulation (regression coefficient of 0.170), surpassing its impact on international circulation (regression coefficient of 0.156). This disparity can be attributed to the fact that the eastern coastal regions are predominantly export-oriented economies, where multimodal transportation plays a more prominent role in international circulation. Meanwhile, the central region, mostly comprising inland cities focused on domestic trade, sees a more noticeable promotional effect of multimodal transportation on domestic circulation.

## 6. Results and policy recommendations

### 6.1. Conclusions of the study

Using panel data from 31 provinces (municipalities and autonomous regions) in China from 2017 to 2023 as the research sample, this study empirically examined the effects of multimodal transportation on domestic and international dual-circulation. The samples were divided into eastern, central, and western regions, and a fixed-effects model was constructed to empirically analyze the impact of multimodal transportation on domestic and international dual-circulation. The main research conclusions are as follows:

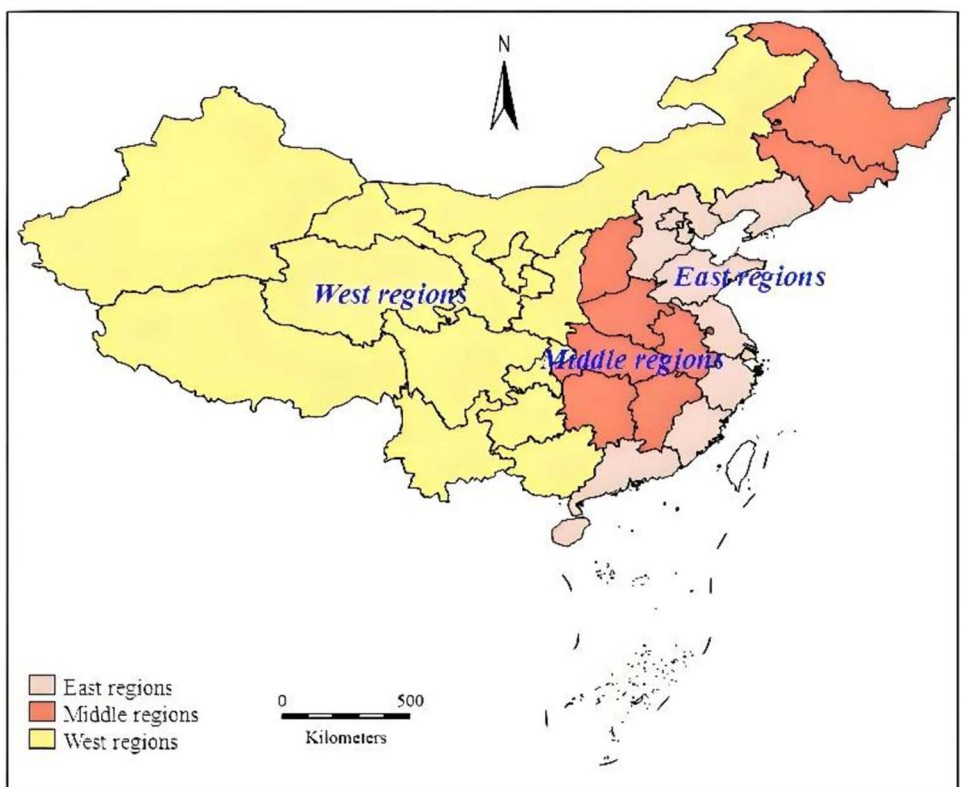

**Fig 2. Provincial distribution map of China's three major economic regions.**

(1) The development of multimodal transportation has positively contributed to both domestic and international dual-circulation, with a more pronounced effect on international circulation.

(2) The influence of multimodal transportation on domestic and international dual-circulation exhibits regional heterogeneity. Specifically, compared to other regions, the development of multimodal transportation in the eastern region exerts a stronger promotional effect on international circulation, while in the central region, it has a more significant impact on domestic circulation. In contrast, the effect of multimodal transportation on domestic and international dual-circulation in the western region is not significant.

(3) Railway multimodal transportation has actively promoted domestic circulation but has a minimal impact on international circulation. On the other hand, waterway multimodal transportation has significantly contributed positively to both domestic and international dual-circulation, with a stronger promotional effect on international circulation.

## 6.2. Policy suggestion

Improve the level of coordination of information services and break down the information barriers between multimodal transport. In multimodal transport, there is the problem of information service coordination, and information sharing between transport enterprises and transport modes cannot be carried out quickly and effectively, resulting in delays and inaccuracies in information transmission. As a result, goods cannot be transmitted and coordinated

in a timely manner between different transportation links, increasing the uncertainty of cargo circulation and the risk of delays. By establishing a unified information platform, the seamless connection of information between different modes of transportation is realized, and the flow of data is unimpeded. In order to effectively connect the various transportation links, it is necessary to develop a set of standardized data interfaces and transmission protocols. Use advanced information technologies such as big data, cloud computing, and the Internet of Things to improve data processing and analysis capabilities to achieve real-time monitoring and intelligent scheduling. Improve the informatization level of employees, strengthen cross-departmental and cross-enterprise collaboration capabilities, form a joint force, and help managers make better decisions, so that the goal of "reducing costs and increasing efficiency" of multimodal transport can be achieved. Improving the level of coordination of information services, breaking down the information barriers between different modes of transportation, and strengthening the interconnection of information systems are one of the important directions for the development of multimodal transport.

Develop multimodal transport according to local conditions and focus on leveraging the comparative advantages of different types of multimodal transport. For the China's eastern coastal regions, where the foundation for multimodal transport development is relatively solid and the degree of openness to the outside world is high, it is suitable to leverage their strengths in water transport, promote the development of multimodal transport towards intelligence and greenness, and encourage port shipping, railway freight, air express, freight forwarding enterprises, and platform-based enterprises to accelerate their transformation into multimodal transport operators. This will further form a development pattern that relies primarily on railways and waterways for the long-distance transport of bulk cargo and containers, thereby strongly supporting the development of international external circulation. For the central and western regions, multimodal transport should leverage the advantages of railways and aviation to accelerate the process of "shifting freight from roads to railways" and "shifting freight from roads to waterways", cultivate market entities for multimodal transport, gradually increase the proportion of multimodal transport in the domestic great circulation, and reduce transportation costs within the domestic great circulation. At the same time, the central and western regions should also actively integrate into the international external circulation, promote interconnection with countries along the "Belt and Road", gradually establish multimodal transport corridors to other countries, and build a smooth and efficient multimodal transport corridor [31].

Strengthen top-level planning and overall management, and focus on improving the combined efficiency of multimodal transport. Due to the transport characteristics of multimodal transport, which involves multiple regions or even different countries, collaboration among different departments is necessary to achieve efficient coordination. In this regard, firstly, the government should strengthen overall planning and establish a time-saving and efficient multimodal transport coordination mechanism among different departments to ensure smooth connections for multimodal transport. Secondly, improve communication and collaboration between government and enterprises, strengthen the ties between local governments and large-scale port and shipping, railway, and logistics enterprises, establish a good communication and coordination mechanism, and actively promote multi-party cooperation for common development. Thirdly, accelerate the establishment of a multimodal transport information platform led by core enterprises or involving multi-enterprise cooperation, to uniformly operate and dispatch all entities on the operation channel, monitor the operation and status of goods throughout the process, conduct comprehensive analysis and coordinated management of intermodal transport information, and respond to emergencies in a timely manner. Fourthly, draw on the experience of transportation integration in coastal and Yangtze

River Delta regions, establish cross-regional intermodal transport coordination mechanisms, promote the integration of road, water, and rail transport among regions, and form a new pattern of dislocation, complementarity, coordination, linkage, joint governance, and sharing.

This paper also has some limitations, leaving room for future research. Firstly, multimodal transport started relatively late in China, with few indicators are available to measure its development level, and the data published by national statistical departments are relatively limited. Therefore, more indicators and data can be explored in future research to enrich the evaluation system of multimodal transport. Secondly, multimodal transport may have a deeper impact on the domestic and international dual-circulation system, and the relationship may not necessarily be a simple linear one. Specific effects can be regressed using other models. In future research, we will pay more attention to the more specific impacts of multimodal transport on the domestic and international dual-circulation system, and further clarify the relationship between them.

## Supporting information

**S1 Data. Raw data.**
(XLSX)

## Author contributions

**Data curation:** xueli zheng, Li Yunhan.

**Funding acquisition:** Liu Wei.

**Methodology:** xueli zheng.

**Software:** xueli zheng.

**Supervision:** Li Xia, Liu Li.

**Writing – original draft:** xueli zheng.

**Writing – review & editing:** Liu Wei.

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
