## [Decision Letter · Decision Letter 0]

26 Nov 2024

PONE-D-24-50174Research on the Impact Effect of Multimodal Transport on Domestic and International Dual Circulation: Evidence from China's Railway and Water TransportPLOS ONE

Dear Dr. zheng, Thank you for submitting your manuscript to PLOS ONE. After careful consideration, we feel that it has merit but does not fully meet PLOS ONE’s publication criteria as it currently stands. Therefore, we invite you to submit a revised version of the manuscript that addresses the points raised during the review process.

**I have completed my evaluation of your manuscript. The reviewers recommend reconsideration of your manuscript following revision and modification. After very careful consideration, I invite you to resubmit your manuscript after addressing the reviewers' comments below.**

We look forward to receiving your revised manuscript.

Kind regards,

Xu Xin

Academic Editor

PLOS ONE

**Journal Requirements:**

Science and Technology project of Henan Provincial Department of Transport "Application of Multimodal transport in Express Logistics" (No. : 2018-2-1), "Research on the Construction of East-bound Multimodal Transport in Henan Province" (No. : 2021G1); Funded by Logistics Research Center, Key Research Base of Humanities and Social Sciences, Henan University, "Research on Policy Support System and Effect Evaluation of Multimodal Transport in China" (No. : 2020-JD-04)

3. In the online submission form, you indicated that your data is available only on request from a third party. Please note that your Data Availability Statement is currently missing the contact details for the third party, such as an email address or a link to where data requests can be made. Please update your statement with the missing information. 

Reviewers' comments:

Reviewer's Responses to Questions

**Comments to the Author**

1. Is the manuscript technically sound, and do the data support the conclusions?

Reviewer #1: Partly

Reviewer #2: Partly

Reviewer #3: Yes

2. Has the statistical analysis been performed appropriately and rigorously? 

Reviewer #1: Yes

Reviewer #2: N/A

Reviewer #3: Yes

3. Have the authors made all data underlying the findings in their manuscript fully available?

Reviewer #1: Yes

Reviewer #2: Yes

Reviewer #3: Yes

4. Is the manuscript presented in an intelligible fashion and written in standard English?

Reviewer #1: Yes

Reviewer #2: Yes

Reviewer #3: Yes

5. Review Comments to the Author

**Reviewer #1:**

This paper analyzed the theory of the multimodal transport's impact on domestic and international dual circulation, provided a practical basis and reference for further research on the development of multimodal transport in relation to domestic and international dual circulation.

The overall work is meaningful. However, there are some drawbacks needing to be improved. The major comments are given below:

(1) It is suggested to add consecutive line numbers.

(2) In terms of literature review, the distinction between the research content of existing references is not clear enough. It is recommended to expand the number of references and try to highlight the innovative points of this paper in tables or other forms.

(3) In section 4.1, the explanation of the parameters and variables in formula (1) does not correspond to the formula.

1) In the subscript, which does i represent, the province or the year?

2) Please explain α0, α1, and α2 separately.

3) β1, β2, Zit, and subsequent symbols do not appear in formula (1).

(4) In section 4.2.1, the text mentions "further open China's coastal areas by developing labor-intensive and export-oriented processing businesses". China's coastal areas are relatively developed, and in recent years, they have been upgrading their industries. Is this statement inappropriate?

(5) Both 4.2.1 and 4.2.2 involve explanatory variables, it is suggested to handle them together.

(6) In section 5.3, the random-effects model is mentioned. Does the random-effects model used in the case have a specific expression? In terms of the dual circulation issue of domestic and international, which model is more suitable, the fixed-effects model or the random-effects model?

(7) In section 5.3, why is tail reduction used to remove extreme data, is it because the data structure is similar to a queue or a stack?

(8) In section 5.4, Table 9 mentions the eastern, central, and western regions. It is suggested to provide the criteria and results for the division of these regions.

(9) In section 6.2, it is suggested to subdivide policy suggestions to correspond to the detailed items in the conclusions of section 6.1, in order to highlight the contributions of the paper.

**Reviewer #2:**

This study has delved into the theoretical mechanism of how multimodal transportation influences both domestic and international dual circulation. However, there are some areas that require improvement. The main comments are as follows:

(1) It is recommended to add more references, summarize the deficiencies in existing research as well as the innovation points of the current study.

(2) Why opt for a fixed effects model? What models are commonly employed in existing research? Compared to existing models, what advantages does your proposed model have?

(3) The explanation of the model symbols does not correspond to the formula. Please provide a detailed explanation of each component in the model and the reasons for selecting these factors.

(4) Could you elucidate the process of regression analysis, bridging the model with the case study in detail?

(5) It is necessary for the manuscript to enumerate the provinces in the eastern, central, and western regions.

(6) Could you discuss the limitations of this study and suggest potential avenues for future research? Are there any areas that need further investigation?

**Reviewer #3:**

This paper empirically tests the mechanism and impact of multimodal transport on domestic and international dual circulation by constructing the fixed-effect model. I have some comments below:

1. In this paper, there are too many long sentences and subordinate clauses, and these sentences should be split for better reading. Moreover, some names, provinces and grammatical expressions are not appropriate, such as "General Secretary Xi Jinping","Shaanxi Province" (Shaanxi Province, China), and "the specific impacts" (the specific impact). It is recommended that the authors make a professional touch-up of the paper.

2. Literature review is too simple, and it can be extend on the importance and impact of multimodal transportation.

3. The review of literature 5 is too brief.

4. The panel data in Table 1 should be extended to the most recent.

5. Regarding Table 2 and 3�the discussion is not enough. How to confirm that the impact of multimodal transport on the domestic circulation remains positive? Why is the impact significant, and the regression results are robust and reliable?”

6. In Section 5.3, regarding the sentence "Drawing on the methodologies of relevant scholars", some appropriate literature should be cited to demonstrate that this is a viable methodology for this paper.

7. In data preprocessing, there is a lack of description of how missing values are handled.

8. I suggest that limitations are discussed in Section 6.1 not 6.2.

6. PLOS authors have the option to publish the peer review history of their article (what does this mean? ). If published, this will include your full peer review and any attached files.

**Do you want your identity to be public for this peer review?** For information about this choice, including consent withdrawal, please see our Privacy Policy .

Reviewer #1: No

Reviewer #2: No

Reviewer #3: No

---

## [Author Response · Author response to Decision Letter 1]

8 Jan 2025

Dear Editor

We are very grateful to your and the reviewers’ critical comments and thoughtful suggestions. Based on these comments and suggestions, we have made careful modification on the original manuscript. All changes made to the text are in red in the revised manuscript so that they may be easily identified. Some of your questions were answered below.

Once again, we acknowledge your comments and constructive suggestions very much, which are valuable in improving the quality of our manuscript.

Here are our responses to the reviewers’ comments one-by-one.

Ⅰ. Reply to the comments of the first review expert

1. It is suggested to add consecutive line numbers.

We added consecutive line numbers.

2. In terms of literature review, the distinction between the research content of existing references is not clear enough. It is recommended to expand the number of references and try to highlight the innovative points of this paper in tables or other forms.

We increased the number of references, made the classification of literature review more complete and clear, and expounded the innovative points of the paper more prominently in the paper. (2 Literature review)

3. In section 4.1, the explanation of the parameters and variables in formula (1) does not correspond to the formula.

1) In the subscript, which does i represent, the province or the year?

In the subscript, i represents the province and t represents the year. We have marked it in the original text.

2) Please explain α0, α1, and α2 separately.

is the intercept term, 、 all are coefficients, that is, the specific impact of multimodal transport on the domestic and international double cycle.

3) β1, β2, Zit, and subsequent symbols do not appear in formula (1).

The three symbols are writing errors. We are sorry for our carelessness and have corrected them in the article.

4. In section 4.2.1, the text mentions "further open China's coastal areas by developing labor-intensive and export-oriented processing businesses". China's coastal areas are relatively developed, and in recent years, they have been upgrading their industries. Is this statement inappropriate?

The statement of "further opening up the coastal areas" has been corrected in the article, and the development of international external circulation can continue to expand the market scale and further introduce foreign investment to develop the economy.

5. Both 4.2.1 and 4.2.2 involve explanatory variables, it is suggested to handle them together.

The variables in section 4.2.1 are explained variables, that is, dependent variables in the regression model; The variable in section 4.2.2 is the core explanatory variable, that is, the independent variable in the regression model. The previous expression was not clear enough and has now been modified.

6. In section 5.3, the random-effects model is mentioned. Does the random-effects model used in the case have a specific expression? In terms of the dual circulation issue of domestic and international, which model is more suitable, the fixed-effects model or the random-effects model?

The expression of the random effects model has been added to the section 5.3 Robustness test. Before choosing whether to use the fixed effects model or the random effects model for regression, we first conducted the Hausmann test. After the test, the experimental data were more suitable for the fixed effects model for regression, so the fixed effects model was chosen as the main regression model in this paper.

7. In section 5.3, why is tail reduction used to remove extreme data, is it because the data structure is similar to a queue or a stack?

There may be some errors in the experimental data we selected, so we choose the method of tail reduction to remove extreme data. The tail reduction carried out in this paper means to find 2.5% and 97.5% of the extreme value of the data, and then replace the number less than 2.5% with the number at 2.5%, and replace the number greater than 97.5% with the number at 97.5%, and the original data is directly changed into the new data.

8. In section 5.4, Table 9 mentions the eastern, central, and western regions. It is suggested to provide the criteria and results for the division of these regions.

According to the documents of the National Development and Reform Commission of China, the research samples were divided into eastern, central and western regions. Among them, eastern and western regions were divided according to the social and economic development status of different regions in China. The eastern region included Beijing, Fujian, Guangdong, Hainan, Hebei, Jiangsu, Liaoning, Shandong, Shanghai, Tianjin and Zhejiang. The central region includes Anhui, Heilongjiang, Henan, Hubei, Hunan, Jiangxi, Jilin and Shanxi, while the western region includes Gansu, Guangxi, Chongqing, Guizhou, Inner Mongolia, Ningxia, Qinghai, Shaanxi, Sichuan, Xinjiang, Yunnan and Tibet. Relevant content has been added in the article.

9. In section 6.2, it is suggested to subdivide policy suggestions to correspond to the detailed items in the conclusions of section 6.1, in order to highlight the contributions of the paper.

The content of policy recommendations has been supplemented according to the opinions of experts, and the conclusion of the paper corresponds to the policy recommendations one by one.

Ⅱ. Reply to the comments of the second review expert

1. It is recommended to add more references, summarize the deficiencies in existing research as well as the innovation points of the current study.

We increased the number of references, made the classification of literature review more complete and clear, and expounded the innovative points of the paper more prominently in the paper. (2 Literature review)

2. Why opt for a fixed effects model? What models are commonly employed in existing research? Compared to existing models, what advantages does your proposed model have?

Before choosing the fixed effects model or the random effects model for regression, we first conducted the Hausmann test. After the test, the experimental data were more suitable for the fixed effects model for regression, so the fixed effects model was chosen as the main regression model in this paper.

3. The explanation of the model symbols does not correspond to the formula. Please provide a detailed explanation of each component in the model and the reasons for selecting these factors.

We have revised and interpreted the symbols of the model, in which, Yit is the explained variable, representing the development level of domestic and international dual-circulation in provinces and years; is the intercept term; 、 are coefficients, represent the specific impacts of multimodal transport on domestic and international dual-circulation; Xit is the core explanatory variable, represents the level of multimodal transport in different provinces and years; represents control variables, including regional economic development level, government support, employment population, and per capita disposable income; represents the provincial effect, represents the year effect, and represents the error term.

4. Could you elucidate the process of regression analysis, bridging the model with the case study in detail?

The processing of model and real case data and the specific links have been added to section 4.1�“The dependent variables of domestic large cycle and international external cycle change with the change of multimodal transport and time, and the statistical data are in units of years. Is the intercept term, is the conditional average when Xit =0, that is, Yit , the development level of domestic and international double circulation when the level of multimodal transport is 0. For the purpose of observation, represents the impact of multimodal transport on the domestic and international double cycle.”

5. It is necessary for the manuscript to enumerate the provinces in the eastern, central, and western regions.

According to the documents of the National Development and Reform Commission of China, the research samples were divided into eastern, central and western regions. Among them, eastern and western regions were divided according to the social and economic development status of different regions in China. The eastern region included Beijing, Fujian, Guangdong, Hainan, Hebei, Jiangsu, Liaoning, Shandong, Shanghai, Tianjin and Zhejiang. The central region includes Anhui, Heilongjiang, Henan, Hubei, Hunan, Jiangxi, Jilin and Shanxi, while the western region includes Gansu, Guangxi, Chongqing, Guizhou, Inner Mongolia, Ningxia, Qinghai, Shaanxi, Sichuan, Xinjiang, Yunnan and Tibet. Relevant content has been added in the article.

6. Could you discuss the limitations of this study and suggest potential avenues for future research? Are there any areas that need further investigation?

Based on your comments and those of other reviewers, we have revised the limitations and future research directions of this paper to section 6.1.“This paper also has some limitations, leaving room for future research. Firstly, multimodal transport started relatively late in China, with few indicators are available to measure its development level, and the data published by national statistical departments are relatively limited. Therefore, more indicators and data can be explored in future research to enrich the evaluation system of multimodal transport. Secondly, multimodal transport may have a deeper impact on the domestic and international dual-circulation system, and the relationship may not necessarily be a simple linear one. Specific effects can be regressed using other models. In future research, we will pay more attention to the more specific impacts of multimodal transport on the domestic and international dual-circulation system, and further clarify the relationship between them.”

Ⅲ. Reply to the comments of the third review expert

1. In this paper, there are too many long sentences and subordinate clauses, and these sentences should be split for better reading. Moreover, some names, provinces and grammatical expressions are not appropriate, such as "General Secretary Xi Jinping", "Shaanxi Province" (Shaanxi Province, China), and "the specific impacts" (the specific impact). It is recommended that the authors make a professional touch-up of the paper.

For the problems mentioned by experts, we have revised them and added country restrictions to the nouns with regional characteristics.

2. 3.Literature review is too simple, and it can be extend on the importance and impact of multimodal transportation.

We increased the number of references, made the classification of literature review more complete and clear, and expounded the innovative points of the paper more prominently in the paper. (2 Literature review)

4. The panel data in Table 1 should be extended to the most recent.

We updated the experimental data from 2017-2021 to 2017-2023, and re-regressed the model in the paper, and updated the data and results.

5. Regarding Table 2 and 3�the discussion is not enough. How to confirm that the impact of multimodal transport on the domestic circulation remains positive? Why is the impact significant, and the regression results are robust and reliable?”

For Table 2 and Table 3, the analysis is supplemented. For the results presented by regression, *** represents the significance of the regression results, and the positive and negative coefficients represent the positive or negative effects of multimodal transport on the domestic and international double cycle and the degree of effect.

6. In Section 5.3, regarding the sentence "Drawing on the methodologies of relevant scholars", some appropriate literature should be cited to demonstrate that this is a viable methodology for this paper.

In Section 5 and 3, we supplement the literature of the scholars used for reference in the section "learning from the methods of relevant scholars", in which the robustness testing methods such as tail shrinking and regression model replacement refer to the research of scholars such as Hu Hanhui and Gu Xiaoyan.

7. In data preprocessing, there is a lack of description of how missing values are handled.

In this paper, a small amount of missing data is supplemented by interpolation method or mean method. This description has been added to the text.

8. I suggest that limitations are discussed in Section 6.1 not 6.2.

We have adjusted the restrictions and future outlook to section 6.1.

Appended to this letter is our point-by-point response to the comments raised by the reviewers. The comments are reproduced and our responses are given directly afterward in a different bold type.

We would like also to thank you for allowing us to resubmit a revised copy of the manuscript. If there are any other modifications we could make, we would like very much to modify them and we really appreciate your help.

On behalf of all the contributing authors, I would like to express our sincere appreciations of your letter and reviewers’ constructive comments concerning our article entitled “Research on the Impact Effect of Multimodal Transport on Domestic and International Dual Circulation: Evidence from China's Railway and Water Transport”. These comments are all valuable and helpful for improving our article.

Sincerely

2025.1.5

---

## [Decision Letter · Decision Letter 1]

15 Jan 2025

PONE-D-24-50174R1Research on the Impact Effect of Multimodal Transport on Domestic and International Dual Circulation: Evidence from China's Railway and Water TransportPLOS ONE

Dear Dr. zheng,

Thank you for submitting your manuscript to PLOS ONE. After careful consideration, we feel that it has merit but does not fully meet PLOS ONE’s publication criteria as it currently stands. Therefore, we invite you to submit a revised version of the manuscript that addresses the points raised during the review process.

We look forward to receiving your revised manuscript.

Kind regards,

Xu Xin

Academic Editor

PLOS ONE

Journal Requirements:

Reviewers' comments:

Reviewer's Responses to Questions

**Comments to the Author**

1. If the authors have adequately addressed your comments raised in a previous round of review and you feel that this manuscript is now acceptable for publication, you may indicate that here to bypass the “Comments to the Author” section, enter your conflict of interest statement in the “Confidential to Editor” section, and submit your "Accept" recommendation.

Reviewer #1: (No Response)

Reviewer #3: All comments have been addressed

2. Is the manuscript technically sound, and do the data support the conclusions?

Reviewer #1: Yes

Reviewer #3: Yes

3. Has the statistical analysis been performed appropriately and rigorously? 

Reviewer #1: Yes

Reviewer #3: Yes

4. Have the authors made all data underlying the findings in their manuscript fully available?

Reviewer #1: (No Response)

Reviewer #3: Yes

5. Is the manuscript presented in an intelligible fashion and written in standard English?

Reviewer #1: Yes

Reviewer #3: Yes

6. Review Comments to the Author

**Reviewer #1:**

The authors have added and explained the content of the article, but there are still some points that need to be improved.

(1) In the literature review, network design should come first, followed by route optimization, and it is suggested that the order of section 2.1 and section 2.2 should be replaced.

(2) The authors should recheck all the text expressions to avoid the clerical errors, for example, in Line 252, “Shanxi” is written as “Shaanxi”. In Line 275, there is no “、” in English.

(3) In Line 299, “labor-intensive and export-oriented processing businesses” is mentioned. Is there any data to support whether labor-intensive industries should continue to dominate in the eastern part of China in the future? Because this seems to be inconsistent with the perception that labor-intensive industries are generally moving to the central and western regions, or even to other countries, please at least provide an explanation in the response.

(4) In Line 469-473, the authors have made necessary additions to the provinces in the eastern, central and western regions, and it is suggested to summarize them in a table instead of too much text. Meanwhile, it is suggested to color the provinces in the east, central and west regions differently based on the blank map of China, which can significantly increase the readability and intuition.

(5) In the structure of the article, it is suggested that the article's summary and limitations (section 6.1) should be placed after the policy suggestion (section 6.2).

(6) It is suggested that in response letter, add changes to the manuscript (with line numbers), with response to the review comments, in a different color.

**Reviewer #3:**

The authors provided clear responses and corresponding revisions to my review comments. Below are my specific comments and suggestions for the current version:

(1) In this paper, there are still some incorrect expressions about provinces. I hope they can be expressed in a standardized manner, such as "Shaanxi Province in China".

(2) Some basic formats of the article are not standardized. For example, line 173, no space between "al." and [23]; in table 9, prefer "domestic circulation" to "Domestic circulation", as all table headers are lowercase. I expect that authors check the full paper to improve the quality.

7. PLOS authors have the option to publish the peer review history of their article (what does this mean? ). If published, this will include your full peer review and any attached files.

**Do you want your identity to be public for this peer review?** For information about this choice, including consent withdrawal, please see our Privacy Policy .

Reviewer #1: No

Reviewer #3: No

---

## [Author Response · Author response to Decision Letter 2]

7 Feb 2025

Dear Editor

We are very grateful to your and the reviewers’ critical comments and thoughtful suggestions. Based on these comments and suggestions, we have made careful modification on the original manuscript. All changes made to the text are in red in the revised manuscript so that they may be easily identified. Some of your questions were answered below.

Once again, we acknowledge your comments and constructive suggestions very much, which are valuable in improving the quality of our manuscript.

Here are our responses to the reviewers’ comments one-by-one.

Ⅰ. Reply to the comments of the first review expert

1. In the literature review, network design should come first, followed by route optimization, and it is suggested that the order of section 2.1 and section 2.2 should be replaced.

According to your suggestion, we have switched the order of Section 2.1 and Section 2.2. At present, it is 2.1 multimodal transport network design and 2.2 multimodal transport path optimization.

2. The authors should recheck all the text expressions to avoid the clerical errors, for example, in Line 252, “Shanxi” is written as “Shaanxi”. In Line 275, there is no “、” in English.

In line 252, examples are Shaanxi Province, China and the Xi 'an Assembly Center of China-Europe freight trains in Xi 'an, the capital of Shaanxi Province; Punctuation problem in line 275 has been corrected.

3. In Line 299, “labor-intensive and export-oriented processing businesses” is mentioned. Is there any data to support whether labor-intensive industries should continue to dominate in the eastern part of China in the future? Because this seems to be inconsistent with the perception that labor-intensive industries are generally moving to the central and western regions, or even to other countries, please at least provide an explanation in the response.

After reviewing relevant materials, we believe that there is indeed some deficiency in the statement "labor-intensive and export-oriented processing businesses" in line 299, which has been revised.

4. In Line 469-473, the authors have made necessary additions to the provinces in the eastern, central and western regions, and it is suggested to summarize them in a table instead of too much text. Meanwhile, it is suggested to color the provinces in the east, central and west regions differently based on the blank map of China, which can significantly increase the readability and intuition.

In Line 469-473, we have summarized the eastern, central and western provinces in the form of tables, and marked the eastern, central and western provinces with different colors on the basis of the blank map of China, which improves the readability and intuition of the article.

5. In the structure of the article, it is suggested that the article's summary and limitations (section 6.1) should be placed after the policy suggestion (section 6.2).

We have placed the summary and limitations of the article (section 6.1) after the policy recommendations (Section 6.2). (In Line 588-596)

6. It is suggested that in response letter, add changes to the manuscript (with line numbers), with response to the review comments, in a different color.

We adopted the revision mode for the revised part of the article, marked it with different colors, and added the specific line number of the modified content in the reply for your convenience.

Ⅱ. Reply to the comments of the second review expert

1. In this paper, there are still some incorrect expressions about provinces. I hope they can be expressed in a standardized manner, such as "Shaanxi Province in China".

We have fixed the syntax error you mentioned (In Line 252).

2. Some basic formats of the article are not standardized. For example, line 173, no space between "al." and [23]; in table 9, prefer "domestic circulation" to "Domestic circulation", as all table headers are lowercase. I expect that authors check the full paper to improve the quality.

In line 173, we have removed the excess Spaces; "Domestic circulation" in Table 9 (now Table 10) has been amended to "domestic circulation". We checked and corrected all these problems.

Appended to this letter is our point-by-point response to the comments raised by the reviewers. The comments are reproduced and our responses are given directly afterward in a different bold type.

We would like also to thank you for allowing us to resubmit a revised copy of the manuscript. If there are any other modifications we could make, we would like very much to modify them and we really appreciate your help.

On behalf of all the contributing authors, I would like to express our sincere appreciations of your letter and reviewers’ constructive comments concerning our article entitled “Research on the Impact Effect of Multimodal Transport on Domestic and International Dual Circulation: Evidence from China's Railway and Water Transport”. These comments are all valuable and helpful for improving our article.

Sincerely

2025.2.7

---

## [Decision Letter · Decision Letter 2]

12 Feb 2025

Research on the Impact Effect of Multimodal Transport on Domestic and International Dual Circulation: Evidence from China's Railway and Water Transport

PONE-D-24-50174R2

Dear Dr. zheng,

We’re pleased to inform you that your manuscript has been judged scientifically suitable for publication and will be formally accepted for publication once it meets all outstanding technical requirements.

Kind regards,

Xu Xin

Academic Editor

PLOS ONE

Reviewers' comments:

Reviewer's Responses to Questions

**Comments to the Author**

1. If the authors have adequately addressed your comments raised in a previous round of review and you feel that this manuscript is now acceptable for publication, you may indicate that here to bypass the “Comments to the Author” section, enter your conflict of interest statement in the “Confidential to Editor” section, and submit your "Accept" recommendation.

Reviewer #1: All comments have been addressed

Reviewer #3: All comments have been addressed

2. Is the manuscript technically sound, and do the data support the conclusions?

Reviewer #1: Yes

Reviewer #3: Yes

3. Has the statistical analysis been performed appropriately and rigorously? 

Reviewer #1: Yes

Reviewer #3: Yes

4. Have the authors made all data underlying the findings in their manuscript fully available?

Reviewer #1: (No Response)

Reviewer #3: Yes

5. Is the manuscript presented in an intelligible fashion and written in standard English?

Reviewer #1: Yes

Reviewer #3: Yes

6. Review Comments to the Author

**Reviewer #1:**

One more comment: the authors may make some minor revision about their index of figures. Two figures are indexed by "Fig.1". After above revision, I think this version is ready for publication.

**Reviewer #3:**

I have reviewed the revised manuscript and found all comments have been addressed. I recommed it can be accepted for publication.

7. PLOS authors have the option to publish the peer review history of their article (what does this mean? ). If published, this will include your full peer review and any attached files.

**Do you want your identity to be public for this peer review?** For information about this choice, including consent withdrawal, please see our Privacy Policy .

Reviewer #1: No

Reviewer #3: No

---

## [Editor Report · Acceptance letter]

PONE-D-24-50174R2

PLOS ONE

Dear Dr. zheng,

I'm pleased to inform you that your manuscript has been deemed suitable for publication in PLOS ONE. Congratulations! Your manuscript is now being handed over to our production team.

Kind regards,

on behalf of

Dr. Xu Xin

Academic Editor

PLOS ONE